# Sec17/Sec18 act twice, enhancing membrane fusion and then disassembling *cis*-SNARE complexes

Hongki Song[1], Amy Orr[1], Mengtong Duan[2], Alexey J Merz[2,3], William Wickner[1]*

[1]Department of Biochemistry and Cell Biology, Geisel School of Medicine, Hanover, United States; [2]Departments of Biochemistry, University of Washington, Seattle, United States; [3]Department of Physiology and Biophysics, University of Washington, Seattle, United States

**Abstract** At physiological protein levels, the slow HOPS- and SNARE-dependent fusion which occurs upon complete SNARE zippering is stimulated by Sec17 and Sec18:ATP without requiring ATP hydrolysis. To stimulate, Sec17 needs its central residues which bind the 0-layer of the SNARE complex and its N-terminal apolar loop. Adding a transmembrane anchor to the N-terminus of Sec17 bypasses this requirement for apolarity of the Sec17 loop, suggesting that the loop functions for membrane binding rather than to trigger bilayer rearrangement. In contrast, when complete C-terminal SNARE zippering is prevented, fusion strictly requires Sec18 and Sec17, and the Sec17 apolar loop has functions beyond membrane anchoring. Thus Sec17 and Sec18 act twice in the fusion cycle, binding to *trans*-SNARE complexes to accelerate fusion, then hydrolyzing ATP to disassemble *cis*-SNARE complexes.

## Introduction

Membrane fusion is essential for each stage of inter-organelle trafficking. Fusion requires lipids and proteins which are conserved among the exocytic and endocytic organelles, and in organisms from yeast to humans. The Rab/Ypt family GTPases are activated for fusion by nucleotide exchange to their GTP-bound state, then bind effector proteins (*Grosshans et al., 2006*). Some effectors are large multisubunit tethering complexes (*Yu and Hughson, 2010*). Fusion also requires SNARE proteins, which associate with each other in cis (when bound to the same membrane) or in trans (when bound to apposed membranes). SNAREs are defined by their heptad-repeat SNARE domains, which can assemble into a 4-helical coiled coil SNARE complex (*Jahn and Scheller, 2006*). Many SNAREs have a C-terminal hydrophobic membrane anchor. In the SNARE domains of the quaternary SNARE complex, polar aminoacyl residues face the exterior. Though the interior of the 4-SNARE coiled coil is largely comprised of apolar residues, each SNARE has an arginyl (R) or glutaminyl (Q) residue at the center of its SNARE domain, facing each other in the center and interior of the 4-SNARE complex. SNAREs are in 4 conserved families, and SNARE complexes are comprised of one SNARE from each of the R, Qa, Qb, and Qc families (*Fasshauer et al., 1998*). SNARE complex assembly is catalyzed by SM (Sec1-Munc18) family proteins (*Rizo and Südhof, 2012*; *Baker et al., 2015*; *Orr et al., 2017*), and disassembly is catalyzed by the ATP-driven chaperone Sec18 (NSF), aided by its co-chaperone Sec17 (αSNAP) (*Söllner et al., 1993*; *Mayer et al., 1996*; *Ungermann et al., 1998*; *Cipriano et al., 2013*).

Biological membranes are covered in protein, so that mere collision is unlikely to bring their bilayer surfaces into contact for fusion. A tethering factor can bind to Rab GTPases on two membranes as a first step towards fusion (*Hickey and Wickner, 2010*; *Miller et al., 2013*). SM proteins,

*For correspondence: William.T.Wickner@dartmouth.edu

**Competing interests:** The authors declare that no competing interests exist.

which may be bound to a Rab's effectors, can then catalyze *trans*-SNARE complex assembly (reviewed in *Baker and Hughson, 2016*). Four SNARE domains 'zipper' together from N-to C-, drawing the membranes together so closely as to exclude other membrane-surface proteins and bring the lipid polar head groups into direct apposition. The formation of continuous α-helices between the SNARE domains and their trans-membrane anchors has been proposed to stress the bilayers to induce fusion (*Hanson et al., 1997a*, *1997b*; *Stein et al., 2009*), yet yeast vacuole and neuronal synaptic fusion can occur with SNAREs that are prenyl-anchored and lack trans-membrane helices (*Xu et al., 2011*; *Zhou et al., 2013*) and even tethering membranes through DNA with lipid anchors at each end can lead to measurable fusion (*Chan et al., 2008*). Full SNARE zippering may also promote fusion though close apposition of bilayers, and by serving as a platform to bind other proteins which trigger the lipid rearrangements of fusion, such as Sec17 (*Zick et al., 2015*) or synaptotagmin (*Fernández-Chacón et al., 2001*). Thus the full array of catalysts and the energetics of fusion remain unclear. Is fusion largely driven by the energy of complete SNARE domain zippering, by the insertion of SNARE-bound proteins into adjacent membrane bilayers, or both?

We study membrane fusion with vacuoles from *S. cerevisiae* (*Wickner, 2010*). The vacuolar tethering complex HOPS (homotypic fusion and vacuole protein sorting) has 6 subunits (*Seals et al., 2000*), two which bind to the Ypt7:GTP vacuolar Rab (*Brett et al., 2008*; *Plemel et al., 2011*; *Bröcker et al., 2012*) to perform tethering (*Hickey and Wickner, 2010*) and one (Vps33) which is the SM protein of that organelle. Vps33 has direct affinity for the SNARE domains of the vacuolar R-SNARE Nyv1 and for the Qa SNARE Vam3 (*Lobingier and Merz, 2012*; *Baker et al., 2015*), enabling HOPS to catalyze the assembly of a complex of the 4 vacuolar SNAREs (*Baker et al., 2015*; *Orr et al., 2017*). The Qc-SNARE Vam7 lacks an integral membrane anchor and is water-soluble; it binds to the membrane by its affinities for the other SNAREs, for the HOPS subunits Vps16 and Vps18 (*Krämer and Ungermann, 2011*), and for PtdIns(3)P (*Cheever et al., 2001*). Vacuole fusion has been studied in vivo (*Wada et al., 1992*), with the purified organelle (*Wickner, 2010*), and as reconstituted into proteoliposomes with all purified components (*Mima et al., 2008*; *Stroupe et al., 2009*; *Zick and Wickner, 2016*).

While Sec17 and Sec18 disassemble *cis*-SNARE complexes, freeing the SNAREs to assemble in trans, four studies have suggested that Sec17 has a function in the fusion cycle which is unrelated to the disassembly of *cis*-SNARE complexes. *Schwartz and Merz (2009)* found that the fusion of purified vacuoles was lost when complete SNARE zippering was prevented through deletion of 21 amino acyl residues (3 heptad repeats) from the C-terminal and membrane-proximal end of the Qc SNARE Vam7. Fusion arrested by this mutant SNARE could be largely restored by the addition of recombinant Sec17. Though the targets and mechanism of Sec17 action on the organelle were unclear, restoration did not require ATP and was insensitive to antibodies to Sec18. A second study, using purified vacuoles which had docked in vitro, found that almost all *trans*-SNARE complexes bore Sec17 (*Xu et al., 2010*). A third study found that Sec17 and the HOPS SM subunit Vps33 could cooperatively associate with vacuole SNARE complexes (*Lobingier et al., 2014*). A fourth study (*Zick et al., 2015*) examined the effects of Sec17 on HOPS-dependent proteoliposome fusion in the absence of Sec18 and with high SNARE concentrations. When the lipids had optimal, vacuolar head-group composition and fatty acyl fluidity, Sec17 had only a minor stimulatory effect on fusion, but when fusion was strongly suppressed by alternative lipid compositions, fusion became dependent on Sec17 and on its N-domain apolar loop. While this study showed that Sec17 could stimulate *trans*-SNARE paired membranes to fuse in a chemically defined reaction in the complete absence of Sec18, it was unclear whether such stimulation would be seen with physiological lipid composition and SNARE concentrations, since high SNARE levels can bypass and mask many fusion reaction requirements, such as for the Rab and even for HOPS (*Mima et al., 2008*). It has recently been reported (*Zick and Wickner, 2016*) that a physiologically fluid lipid fatty acyl phase allows fusion at dramatically lower SNARE levels. We have therefore re-examined the roles of Sec17, Sec18, and adenine nucleotide under these more physiological conditions, and in assays which do not require *cis*-SNARE complex disassembly.

Though HOPS, Ypt7 and SNAREs can catalyze slow fusion at physiological protein levels and lipid compositions, we now report that this fusion is stimulated by Sec17 and Sec18:ATP without a need for ATP hydrolysis. The apolar loop of the Sec17 N-domain is needed for its membrane binding, and Sec17 interacts with SNAREs to stimulate fusion. We also sought conditions where fusion strictly requires Sec17/Sec18/ATP rather than being simply stimulated by them. Fusion is blocked when the

completion of SNARE domain zippering is prevented by deletion of 21 residues (3 heptad repeats) from the Qc-SNARE C-terminus, but can be restored in a reaction that strictly requires Sec17, Sec18, and ATPγS. In this fusion reaction, the Sec17 apolar loop has functions in addition to membrane-binding.

## Results

Our working model of the vacuolar proteins assembled for membrane fusion (*Zick et al., 2015*) is shown in *Figure 1*, based on the structure of the 20 s complex of αSNAPs (Sec17), 4-SNARE domain complex, and NSF (*Zhao et al., 2015*) and on prior studies of vacuole fusion (*Wickner, 2010*; *Wickner and Rizo, 2017*) and its stimulation by Sec17 (*Zick et al., 2015*). This model is consistent with the affinity of two HOPS subunits for Ypt7 and their capacity to mediate tethering, the binding affinity of HOPS for several SNAREs and its catalysis of SNARE complex assembly (*Plemel et al., 2011*; *Baker et al., 2015*; *Orr et al., 2017*), and the enrichment of each of these components at the vacuolar vertex ring microdomain where fusion occurs (*Wang et al., 2002*, *2003*; *Fratti et al., 2004*). Sec17 and SNAREs are at the center. Sec17 has direct affinity for Sec18 and SNAREs (*Cipriano et al., 2013*) and can exhibit SNARE-dependent association with lipid bilayers (*Zick et al., 2015*). To analyze the roles of each of these interacting proteins at the membrane fusion microdomain, we employed a functional assay of fusion (*Zucchi and Zick, 2011*; *Zick and Wickner, 2016*).

Proteoliposomes bearing prenylated Ypt7 at a 1:8000 molar ratio to lipids and SNAREs at 1:32,000 molar ratios to lipids were prepared with lipids of vacuolar head group composition and

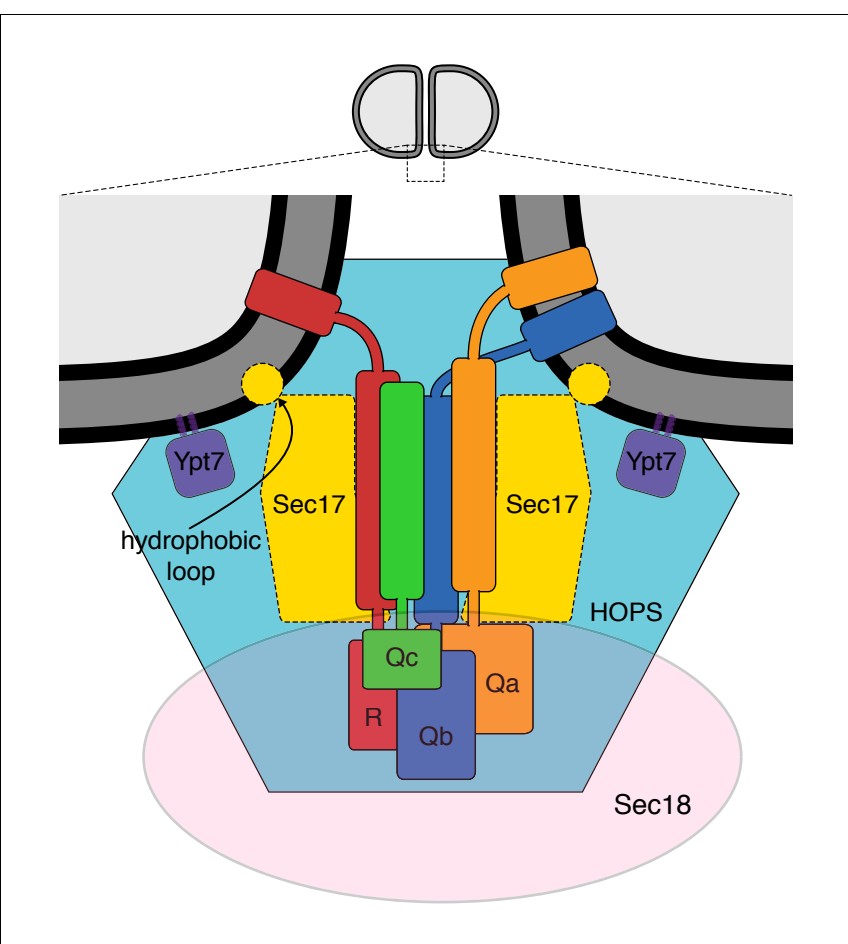

**Figure 1.** The multiple interactions of proteins and membranes for fusion. A working model of the proteins and membranes as assembled prior to vacuole or proteoliposome fusion. From *Zick et al. (2015)*.

fatty acyl fluidity (*Zick and Wickner, 2016*). Proteoliposomes bore entrapped fluorescent proteins to allow assay of fusion (*Zucchi and Zick, 2011*). One set of proteoliposomes bore lumenal Cy5-derivatized streptavidin, while the other had lumenal biotinylated phycoerythrin. When these proteoliposomes are mixed, the entrapped fluorescent proteins are separated by at least the thickness of two bilayers and thus have negligible FRET (fluorescence resonance energy transfer). Upon fusion and the attendant lumenal compartment mixing, there is tight binding of the biotinylated phycoerythrin to Cy5-streptavidin and a strong FRET signal between the two fluorophores. Fusion assays are performed in the presence of a large molar excess of external nonfluorescent streptavidin, capturing any extralumenal biotinylated phycoerythrin and thus blocking any FRET signal due to lysis. Proteoliposomes bear the lumenal fluorescent proteins at comparable levels. Most of these fluorescent proteins will bind to each other after a single fusion event, though the proteoliposomes have significant size heterogeneity, yielding unreacted lumenal reporter protein after one round of fusion and considerable signal from subsequent fusion rounds. To examine an initial round of fusion without the need for prior recycling of *cis*-SNARE complexes, two batches of proteoliposomes were prepared, with Ypt7 on each and with either the R-SNARE Nyv1 or with both the Qa and Qb SNAREs, Vam3 and Vti1. At the physiological Rab and SNARE levels employed, the rates of fusion (*Zick and Wickner, 2016*) resemble those seen with purified vacuoles (*Haas et al., 1994*; *Nichols et al., 1997*).

## Fusion activation by Sec17/Sec18 without ATP hydrolysis

When mixed with the soluble Qc-SNARE, proteoliposomes bearing either Ypt7:GTP and R-SNARE or Ypt7:GTP and QaQb-SNAREs undergo slow HOPS-dependent fusion (*Figure 2A*, filled vs open diamonds). Fusion is stimulated by the addition of Sec17, Sec18, and ATP (filled circles). This stimulation might be ascribed to the well known capacity of Sec18/NSF and Sec17/αSNAP to disassemble SNARE complexes, perhaps reflecting some nonproductive complexes of reconstituted SNAREs which would have to be disassembled in order to form functional associations on these proteoliposomes. It is therefore noteworthy that fusion for approximately the first 10 min was almost as rapid with Sec17, Sec18, and ATPγS (*Figure 2A*, open circles; supplementary figures show average and standard deviations) as with ATP (filled circles). The additional fusion at later times with ATP may be due to the recycling of post-fusion *cis*-SNARE complexes, allowing additional rounds of fusion. In contrast to the capacity of ATPγS to support Sec17/Sec18 stimulation of the fusion of R- and QaQb-proteoliposomes, hydrolyzable ATP is strictly required for the fusion of proteoliposomes prepared with all 4 SNAREs in cis-complexes (*Figure 2B*). While ATP or ATPγS had no effect on the fusion mediated by HOPS alone (data not shown), Sec17 gave a small stimulation (*Figure 2A*, filled triangles), as reported (*Zick et al., 2015*). Sec18 without Sec17 also stimulated HOPS-dependent fusion (open triangles), and there was greater stimulation by Sec18 in the presence of ATP (filled squares) or ATPγS (open squares). Further studies will be needed to learn how Sec18 performs this Sec17-independent fusion stimulation. Optimal initial rates of fusion required Sec17, Sec18, and ATP or ATPγS.

To better understand how Sec17 stimulates fusion when not cooperating with Sec18 and hydrolyzable ATP to disassemble SNARE complexes, we performed fusion reactions with Sec18:ATPγS and with Sec17 which was either wild-type or bore functionally targeted mutations. Elegant structural studies of the human neuronal fusion proteins (*Zhao et al., 2015*) have shown that several Sec17 molecules can bind along a bundle of 4 SNARE helices with the C-terminus of each Sec17 near the membrane-distal end of the coiled SNARE domains, where Sec18 also binds to the SNARE bundle, with the central region of Sec17 in contact with the SNAREs, and with the Sec17 N-terminal region near the C-termini of the SNARE domains, immediately adjacent to the bilayer and where the Sec17 apolar loop may insert into lipid (*Figure 1*). Mutations within αSNAP/Sec17 allow tests of the functional importance of each of these Sec17 interactions. The F21S,M22S mutation of Sec17 removes the apolar character of a loop of Sec17 near its N-terminus (*Winter et al., 2009*), the L291A,L292A mutation near the C-terminus of Sec17 alters its interaction with Sec18 (*Barnard et al., 1997*; *Schwartz and Merz, 2009*), and a conserved lysyl residue at position 163 in αSNAP (corresponding to residue 159 and/or 163 in Sec17) is positioned near the SNARE complex zero-layer where it is important for functional interactions (*Zhao et al., 2015*). In an earlier study at higher SNARE concentrations, Sec17-F21S,M22S was found to be fully-functional for supporting Sec18-mediated SNARE complex disassembly for the fusion of 4-SNARE proteoliposomes, but did not stimulate fusion of R- and Q-SNARE proteoliposomes in the absence of Sec18 (*Zick et al., 2015*).

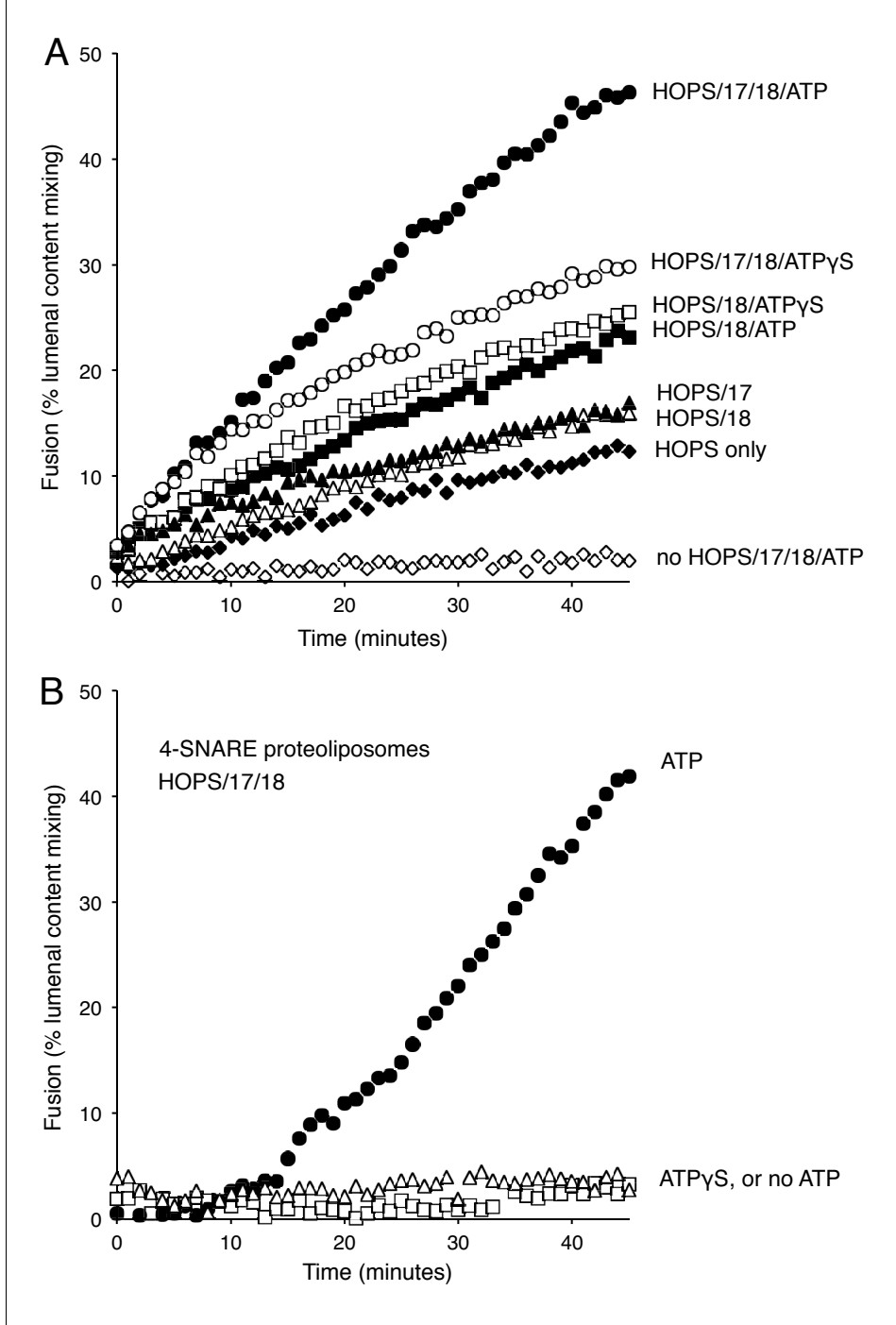

**Figure 2.** Stimulation of fusion by Sec18 does not require ATP hydrolysis, while Sec18-mediated disassembly of cis-SNARE complexes requires ATP hydrolysis. (**A**) The fusion of Ypt7(GTP):R-SNARE proteoliposomes with Ypt7(GTP):QaQb-SNARE proteoliposomes with added Qc-SNARE and HOPS is stimulated by Sec17, Sec18, and ATP or ATPγS. (**B**) ATP hydrolysis is required for 4-SNARE proteoliposome fusion. Proteoliposomes bearing Ypt7:GTP, each of the four vacuolar SNAREs at a 1:32,000 molar ratio to lipids, and either Cy5-streptavidin or biotinylated-phycoerythrin were prepared, mixed, and assayed for fusion in the presence of HOPS, Sec17, Sec18 and ATP (filled circles), ATPγS (open squares), or without addition of adenine nucleotide (open triangles), as described in Materials and Methods. The data shown in this and the following figures were typical of 3 repeat experiments; means and standard deviations for each experiment are presented in the corresponding Supplementary Figures.

*Figure 2 continued on next page*

*Figure 2 continued*

The following source data and figure supplement are available for figure 2:

**Source data 1.** Source data file (Excel) for *Figure 2A*.
**Source data 2.** Source data file (Excel) for *Figure 2B*.
**Figure supplement 1.** Average and standard deviations of fusion after 30 min for triplicate assays as in *Figure 2*, relative to the maximal fusion condition.

The opposite was true of Sec17-L291A,L292A; it was inactive in helping Sec18 to promote 4-SNARE proteoliposome fusion, but was fully active in promoting Sec18-independent fusion of either vacuoles or proteoliposomes (*Schwartz and Merz, 2009*; *Zick et al., 2015*). Model studies with liposomes have shown that wild-type Sec17 can interact cooperatively with a mixture of vacuolar SNAREs without their membrane anchors (termed 'soluble SNAREs,' or sSNAREs), forming a 4sSNARE:oligo-Sec17 complex which binds to liposomes far more efficiently than either the sSNAREs alone or the Sec17 alone (Figure 7 of *Zick et al., 2015*). Using this assay, we find that Sec17-K159E,K163E, an analog of the K122E,K163E mutant of α-SNAP (*Marz et al., 2003*), has less capacity to synergistically associate with SNAREs (*Figure 3—figure supplement 2*, lane 8) and be bound to liposomes than wild-type Sec17 (lane 2) or Sec17-L291A,L292A (lane 6). These three Sec17 mutants provide a means to examine the roles of the respective domains of Sec17 in fusion.

To explore fusion without SNARE recycling, incubations were performed with Ypt7(GTP):R-SNARE proteoliposomes and Ypt7(GTP):QaQb-SNARE proteoliposomes at physiological protein: lipid ratios (*Zick and Wickner, 2016*) and with HOPS, various combinations of Sec18 and wild-type or mutant forms of Sec17, and ATPγS (*Figure 3A*). The fusion without Sec17 or Sec18 (solid triangles) was stimulated by Sec17 (filled squares), but there was greater stimulation by Sec18 (filled circles), as above (*Figure 2A*). The combined addition of Sec17 and Sec18 gave stimulation that was approximately additive (*Figure 3A*, open circles), and this absence of synergy was reinforced by the insensitivity of fusion to the Sec17-L291A,L292A mutation (open diamonds) which interferes with functional Sec17:Sec18 interactions. Substitution of Sec17-F21SM22S or Sec17-K159E,K163E instead of wild-type Sec17 (open squares or open triangles, respectively) reduced fusion to the levels seen with without any added Sec17 (solid circles), indicating that the Sec17 N-domain apolar loop and Sec17:SNARE interactions are required.

In the same experiment, parallel fusion incubations were performed with [hydrolyzable] ATP (*Figure 3B*). ATP can fulfill both the activating ligand function seen with ATPγS and the SNARE-recycling functions whereby Sec17 and Sec18 support later fusion cycles. With added HOPS, Vam7, and ATP, fusion was seen without Sec17 or Sec18 (solid triangles), stimulated by Sec18 alone (solid circles), Sec17 alone (solid squares), or Sec18 and Sec17 L291A,L292A (open diamonds), which blocks Sec17:Sec18 interaction for the cis-SNARE complex disassembly which would otherwise allow subsequent fusion rounds. The fusion seen in the presence of ATP was in each of these cases similar to that seen with non-hydrolyzable ATPγS (*Figure 3—figure supplement 3*; black symbols indicate fusion with ATP, blue symbols indicate fusion with ATPγS). However, ATP and wild-type Sec17 and Sec18 supported more extensive fusion at later incubation times (*Figure 3B*, open circles), presumably reflecting the recycling of cis-SNARE complexes after fusion. There was only partial diminution of this fusion with Sec17-F21S,M22S with its attendant loss of the N-loop apolarity (open squares) or with Sec17-K159E,K163E with its diminished SNARE affinity (open triangles). A direct comparison (*Figure 3—figure supplement 4*) shows very similar diminution of fusion by the K159E,K163E and F21S,M22S mutations in incubations with ATP (black symbols) or with ATPγS (blue symbols), and thus it is likely that these mutations only affect the first fusion round, which is all that ATPγS can support. ATP-dependent SNARE recycling stimulates fusion over that seen with ATPγS, whether with wild-type Sec17, Sec17-K159E,K163E or Sec17-F21S,M22S (*Figure 3—figure supplement 4*), indicating that Sec17 can support Sec18 and ATP-dependent SNARE recycling without either the apolar Sec17 loop or Sec17 interaction with the SNARE 0-layer. Though each of these interactions has been reported to be important for the disassembly of isolated neuronal SNARE complexes (*Marz et al., 2003*; *Winter et al., 2009*; *Zhao et al., 2015*), they may be less vital for the disassembly of vacuolar

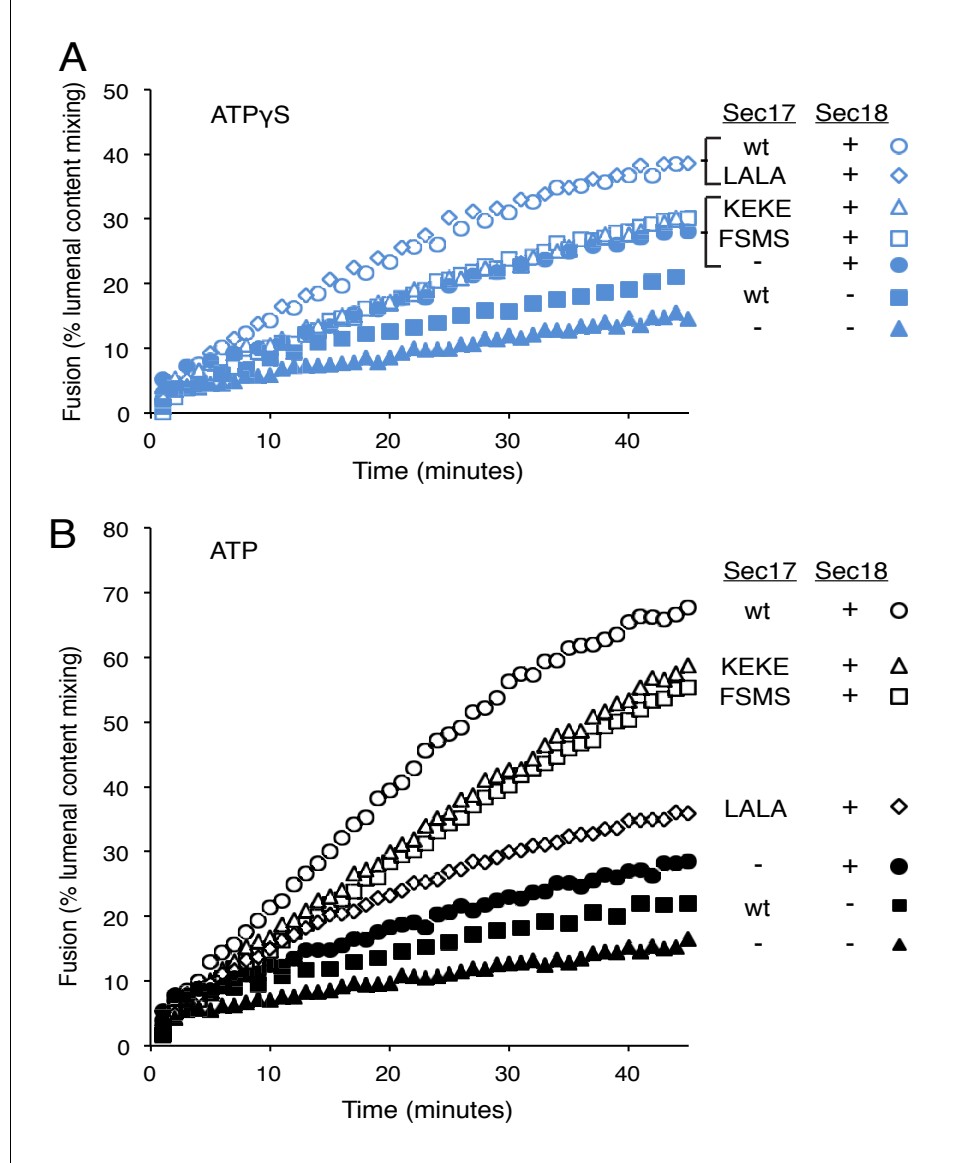

**Figure 3.** Fusion of Ypt7(GTP):R-SNARE and Ypt7(GTP):QaQb-SNARE proteoliposomes in the presence of HOPS, Qc, ATP or ATPγS, and Sec17 (wild type or mutants) and/or Sec18, as indicated.  Fusion conditions were as described in Materials and methods. (**A**) Fusion with ATPγS. (**B**) Fusion with ATP. Sec17 was wild type (wt) or had the mutations F21S,M22S (FSMS), K159E,K163E (KEKE), or L291A,L292A (LALA). These data are from the same experiment.

The following source data and figure supplements are available for figure 3:

**Source data 1.** Source data file (Excel) for *Figure 3* Parts A, B, and C.

**Figure supplement 1.** Average and standard deviations of fusion after 30 min for triplicate assays as in *Figure 3*, relative to the maximal fusion condition.

**Figure supplement 2.** The assembly of a complex of soluble vacuolar SNAREs with Sec17 allows both the sSNAREs and the Sec17 to bind to liposomes (*Zick et al., 2015*).

**Figure supplement 3.** Without productive interaction between Sec17 and Sec18, there are equivalent fusion kinetics in the presence of ATP or ATPγS.

*Figure 3 continued on next page*

*Figure 3 continued*

**Figure supplement 4.** The Sec17 K159E,K163E or F21S,M22S mutations diminish fusion to the same extent in incubations with ATP or ATPγS, and there is similar enhancement of fusion by hydrolysable ATP for each Sec17, wild-type or mutants.

membrane-bound SNAREs in the presence of HOPS, with its capacity to stabilize interactions through its multiple affinities.

## Membrane-anchored Sec17

Does membrane insertion of the apolar loop in the N-domain of Sec17 function as a wedge to perturb bilayer structure and trigger fusion, or as one of the binding elements which brings Sec17 to the active fusion microdomain where it can bind to SNARE complexes and stabilize their fully zippered structure (*Ma et al., 2016*), or both? A strictly membrane-binding function might be bypassed by a synthetic transmembrane anchor, which itself would not be expected to disturb the bilayer. We therefore fused the trans-membrane (TM) anchor of the vacuolar Qb SNARE Vti1 to the N-terminus of Sec17 or Sec17-F21S,M22S to create TM-Sec17 and TM-Sec17-F21S,M22S and purified each of the resulting recombinant fusion proteins. These were each mixed in detergent with lipids, the Rab Ypt7 and the R or Qa,Qb SNAREs and detergent was removed by dialysis to prepare TM-Sec17, Ypt7, 1R- or TM-Sec17, Ypt7,QaQb-proteoliposomes, which were isolated by floatation. The fusion seen with HOPS but without soluble or membrane-anchored Sec17 (*Figure 4A*, triangles) showed a small stimulation by Sec18 (squares), especially in the presence of ATP or ATPγS (open and filled circles). In the presence of ATP or ATPγS, fusion was stimulated by TM-Sec17 (*Figure 4B*). TM-Sec17-F21S,M22S showed a similar stimulation of fusion activity (*Figure 4C* and *Figure 4—figure supplement 1*), in contrast to the findings with Sec17 F21SM22S that is not membrane anchored (*Figure 3A*). These findings suggest that the stimulation of fusion by Sec 17 in the presence of wild-type SNAREs, HOPS, and Sec18:ATP largely relies on the Sec17 apolar loop for membrane affinity rather than for a bilayer-restructuring wedge function.

## Fusion without full, C-terminal SNARE zippering

Since fusion with HOPS and wild-type vacuolar SNAREs is only stimulated by Sec17 and Sec18, we sought conditions where Sec17, Sec18, and ATP or ATPγS would be strictly required for fusion. Fusion is blocked by Qc-3Δ, a well-studied mutant form of the vacuolar Qc-SNARE Vam7 lacking the 21 C-terminal aminoacyl residues of its SNARE domain (*Schwartz and Merz, 2009*). While reconstituted proteoliposomes fused when incubated with wild-type Qc, HOPS, Sec17, Sec18, and ATP (*Figure 5A*, filled circles), substitution of Qc-3Δ (filled squares) prevented fusion as completely as the omission of HOPS (filled triangles). However, despite the arrest to the completion of SNARE zippering, a striking level of fusion was preserved in incubations with Qc-3Δ when ATPγS was present (open circles vs open squares).

The proteoliposome fusion which occurs with Qc-3Δ and ATPγS in the absence of complete C-terminal SNARE zippering requires both Sec17 and Sec18 (*Figure 5B*, filled circles, vs. filled squares and triangles), in clear contrast to fusion with wild-type Qc (open circles) where complete SNARE zippering can occur and Sec17 and Sec18 are stimulatory but not essential (open triangles and squares). Limited fusion is seen with higher concentrations of Sec17 alone (data not shown), suggesting that one important function of Sec18 is to stabilize the binding of Sec17. ATPγS has a positive role in this reaction, and is not simply acting to block the hydrolysis of any low levels of remaining ATP, as the absence of added adenine nucleotide and complete removal of any residual ATP by glucose/hexokinase does not allow fusion in the presence of Qc-3Δ (*Figure 5C*, filled symbols).

How is fusion being stimulated by Sec17 in the absence of C-terminal SNARE domain zippering? We addressed this with the same Sec17 mutants which either affect its hydrophobic interactions with membranes near the Sec17 N-terminus, with the SNARE 0-layer near the middle of Sec17, and with Sec18 near the Sec17 C-terminus. Fusion with Qc-3Δ, Sec17, Sec18, and ATPγS (*Figure 6A*, open circles) is markedly diminished by the Sec17-L291A,L292A mutation, altering Sec17:Sec18 interactions (open diamonds), and is abolished by the Sec17-F21SM22S mutation which removes the apolar

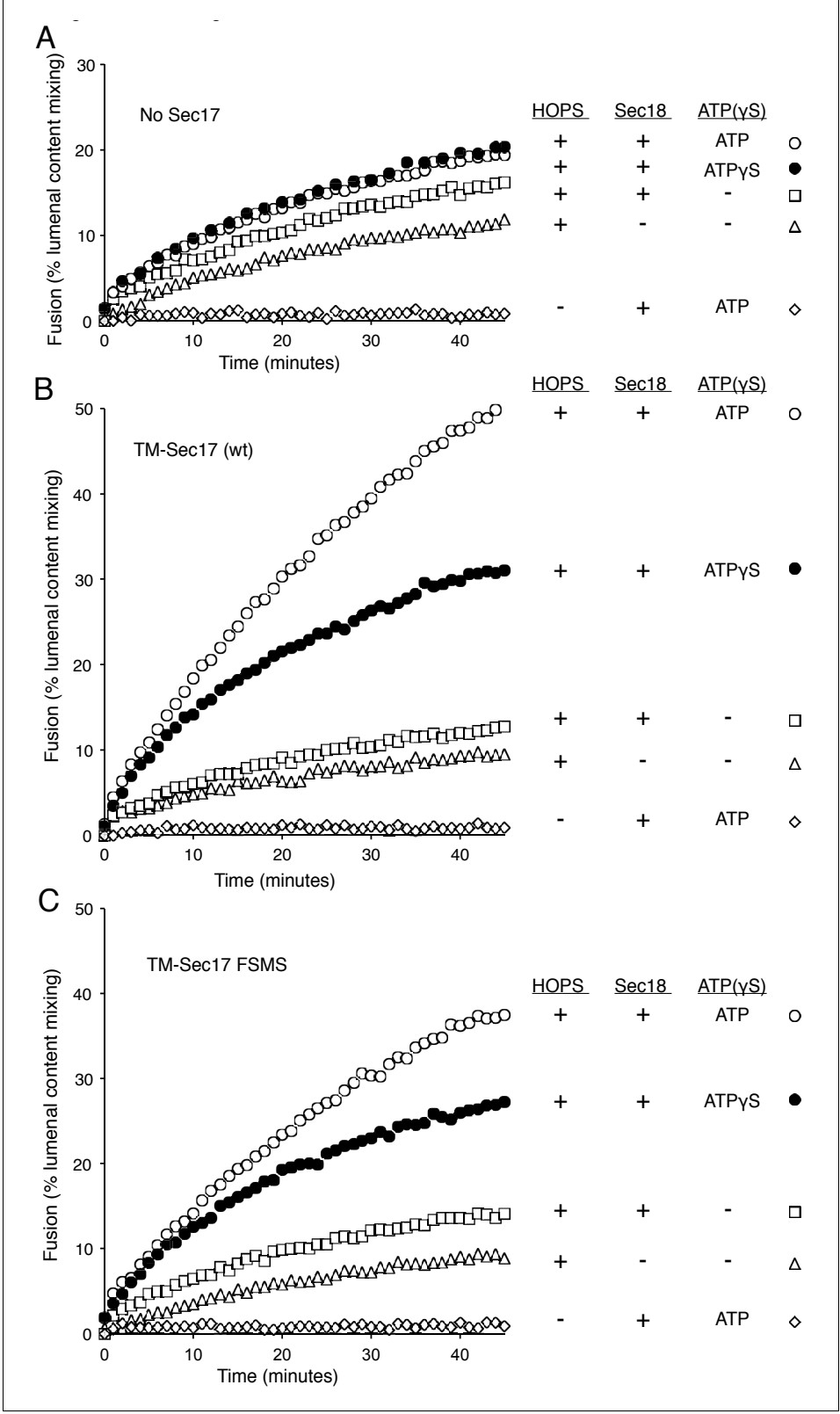

**Figure 4.** Membrane-anchoring Sec17 bypasses the requirement for the apolarity of its N-domain loop. The transmembrane domain from the Qb-SNARE Vti1 was joined to the N-terminus of Sec17 (wt) or Sec17 F21S,M22S (FSMS) to give TM-Sec17 (wt) and TM-Sec17 FSMS. (**A**) No Sec17, (**B**) TM-Sec17 (wt), or **C**). TM-Sec17 FSMS were included at a 1:8000 molar ratio to lipids in the reconstitution of Ypt7(GTP):R-SNARE and Ypt7(GTP):QaQb-SNARE

*Figure 4 continued on next page*

*Figure 4 continued*

proteoliposomes (Ypt7 and each SNARE were added at 1:8000 and 1:40,000 molar ratios to lipids, respectively). Fusion incubations with these proteoliposomes had HOPS, Qc, Sec18, and no ATP, ATP, or ATPγS as indicated.

The following source data and figure supplement are available for figure 4:

**Source data 1.** Source data file (Excel) for *Figure 4* Parts A, B, and C.

**Figure supplement 1.** Average and standard deviations of fusion after 30 min for triplicate assays as in *Figure 4*, relative to the maximal fusion condition.

character of the Sec17 N-domain loop (open squares), or by the Sec17-K159E,K163E mutant which interferes with SNARE interactions (open triangles). With Qc-3Δ, no fusion was seen at all with hydrolysable ATP (*Figure 5A*, *Figure 6B*). Fusion with Qc-3Δ was also examined with membrane anchored TM-Sec17 or TM-Sec17 F21SM22S. Fusion with TM-Sec17 (*Figure 6C*, open circles) was diminished by the FSMS mutation (open squares). Since the fusion was quite limited with TM-Sec17 and Qc-3Δ, we re-optimized the reactions and found enhanced fusion with 100 nM HOPS instead of 32 nM HOPS. The more rapid fusion seen under this condition (filled circles) was strongly diminished by the FSMS mutation (filled squares). These data indicate that the Sec17 apolar loop provides more than a membrane-anchoring function in the absence of complete SNARE zippering.

## Staging Sec17-triggered fusion

To resolve the formation of competent intermediates from fusion per se, we incubated Ypt7(GTP):R-SNARE proteoliposomes with Ypt7(GTP):QaQb-SNARE proteoliposomes in the presence of either HOPS, Sec17, Sec18, ATPγS, and Qc-3Δ (*Figure 7*, open circles, no omission) or with one or two of these components omitted during the initial 20 min of incubation but then added back. Omission of Sec17 prevented fusion, but there was a rapid burst of fusion upon its re-addition (open triangles), faster than the no-omission control, indicating that a productive intermediate had formed. Omission of Sec17 plus any other reaction component (Sec18, filled triangles; Qc-3Δ, filled squares; ATPγS, filled circles; HOPS, filled diamonds) and the restoration of both to the reactions after 20 min allowed reconstitution of the initial rate of fusion (open circles), but not at the rate or extent seen when only Sec17 had been absent initially (open triangles). These data suggest that the assembled but partially-zippered trans-SNARE complex with bound Sec18:ATPγS formed an active site for Sec17 to bind and trigger rapid fusion.

## Discussion

We have explored the roles of Sec17, Sec18, and ATP, in fusion as well as in SNARE complex recycling. The functions of Sec17 and Sec18, and their adenine nucleotide requirement, are distinct when the *trans*-SNAREs are fully zippered, when zippering is incomplete, or after fusion when *cis*-SNARE complexes must be disassembled to permit subsequent fusion rounds. *Cis*-SNARE complex disassembly requires ATP hydrolysis and Sec17:Sec18 interactions which are sensitive to the Sec17 L291A,L292A mutation. In contrast, an initial round of fusion with wild-type and fully-zippered SNAREs requires only HOPS and the SNAREs, but is stimulated by Sec17 and Sec18 in a distinct manner, requiring ATP without a need for ATP hydrolysis, requiring the Sec17 apolar loop for membrane binding, and needing Sec17:SNARE association for full fusion stimulation. When SNARE zippering is incomplete, the relief of the fusion blockade strictly requires Sec17, Sec18, and a nonhydrolyzable ATP analog and is sensitive to mutations which interfere with the Sec17 N-loop apolarity, with Sec17:SNARE 0-layer interaction, and with Sec17:Sec18 interaction. Without complete SNARE zippering, membrane-anchoring Sec17 by a recombinant trans-membrane domain does not relieve the need for the N-loop apolarity, suggesting that this apolar loop fulfills an additional function. Together these findings establish that Sec17 and Sec18 act during fusion in a distinct manner from their ATP hydrolysis-dependent disassembly of post-fusion cis-SNARE complexes.

At least two nonexclusive mechanisms may contribute to Sec17 stimulation of fusion. In one, a 'Wedge Model' (*Zick et al., 2015*), Sec17 is bound and positioned by its associations with both

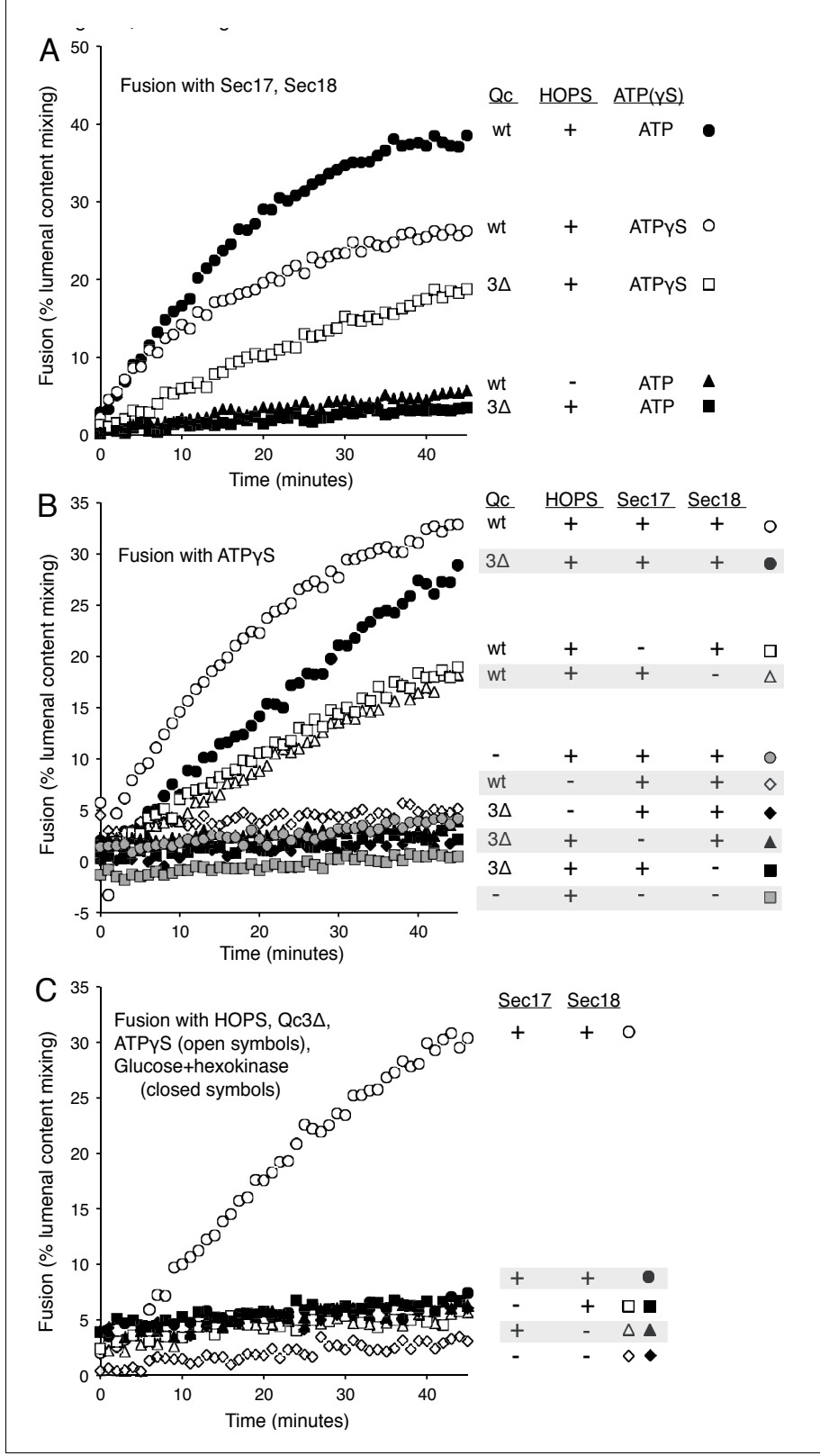

**Figure 5.** Requirements for fusion in the absence of C-terminal SNARE domain zippering. (**A**) Fusion is lost without SNARE zippering unless ATP is replaced by ATPγS. The fusion of proteoliposomes with Ypt7:GTP and either the R-SNARE or QaQb-SNAREs in the presence of Sec17 and Sec18 was assayed with ATP or ATPγS, with wild-type Vam7 or Vam7-3Δ and with or without HOPS, as indicated, as described in Materials and Methods. The

*Figure 5 continued on next page*

*Figure 5 continued*

final concentration of $MgCl_2$ was 0.69 mM. (**B**) With incomplete SNARE zippering, fusion requires HOPS, Sec17, and Sec18. Fusion reactions had ATPγS, and either wild-type Qc or Qc-3Δ as well as Sec17, Sec18, and HOPS. Fusion reactions with wild-type Qc are depicted with open symbols, those with Qc-3Δ are filled symbols. Fusion with Vam7-3Δ was lost upon omission of HOPS (filled black diamonds), Sec17 (solid triangles), or Sec18 (solid squares). Fusion reactions without any Qc are in gray symbols, in the presence of HOPS, Sec17 and Sec18 (circles) or without Sec17/Sec18 (squares), as described in Materials and methods. (**C**) Arrested SNARE zippering requires Sec17, Sec18, and ATPγS for fusion. Fusion of Ypt7(GTP):R-SNARE proteoliposomes with Ypt7(GTP):QaQb-SNARE proteoliposomes was performed as described in Materials and Methods, but with HOPS and the C-terminally truncated Qc SNARE Vam7-3Δ and with either ATPγS (open symbols) or glucose/hexokinase (filled symbols). Incubations had Sec17 and Sec18 (circles), Sec18 (squares), Sec17 (triangles), or neither (diamonds). The final concentration of $MgCl_2$ was 0.69 mM.

The following source data and figure supplement are available for figure 5:

**Source data 1.** Source data file (Excel) for *Figure 5A*.
**Source data 2.** Source data file (Excel) for *Figure 5B*.
**Source data 3.** Source data file (Excel) for *Figure 5C*.
**Figure supplement 1.** Average and standard deviations of fusion after 30 min for triplicate assays as in *Figure 5*, relative to the maximal fusion condition.

---

Sec18 and the *trans*-SNARE complex. It may then trigger fusion by insertion of its hydrophobic loop into the adjacent monolayer of either or both of the two apposed bilayers. When C-terminal zippering is incomplete (*Figure 6C*), TM-Sec17 supports fusion far better than TM-Sec17-F21S,M22S, indicating that the apolar loop is not simply needed for membrane binding, but might act as a wedge to perturb the bilayers. However, with full SNARE zippering, the addition of a membrane anchor to Sec17 bypasses the need for the apolar character of its loop (*Figure 4*), suggesting that the loop normally promotes Sec17 membrane association.

A second, '*trans*-SNARE Stability Model' of Sec17 function postulates that the complete zippering of vacuolar SNAREs in trans is energetically strained and readily reverses to a partially-zippered state (*Schwartz and Merz, 2009*; *Ma et al., 2016*). This would cause a low steady-state abundance of completely zippered SNAREs, consistent with the observed slow fusion rate with HOPS and SNAREs alone (*Figures 2* and *3*). In this model, Sec17, bound and positioned by both its N-domain apolar loop insertion into the bilayer and its association with the fully-zippered *trans*-SNAREs (*Ma et al., 2016*), would promote fusion by keeping a greater portion of the *trans*-SNAREs fully zippered. In accord with this model, *Winter et al. (2009)* found that the membrane association of recombinant neuronal SNAREs strongly enhances α-SNAP/Sec17 and NSF/Sec18 mediated disassembly, and this enhancement relies on the apolar character of the α-SNAP N-domain loop. In these studies, the role of the α-SNAP apolar loop in SNARE complex disassembly is unlikely to require any wedge-like bilayer disruption, but rather reflects a role for this loop in enhancing α-SNAP binding to membrane-bound SNAREs, positioning it to interact with the C-terminal region of the SNARE complex (*Ryu et al., 2015*; *Choi et al., 2016*), where it can initiate NSF/Sec18-driven disassembly. In experiments with either vacuoles or proteoliposomes which cannot complete SNARE zippering due to the Qc-3Δ truncation, the trans-SNARE stability model suggests that Sec17 might stabilize the remaining 3 SNARE domain C-terminal regions in a conformation that resembles the wild-type fully zipped conformation. When the wild-type full-length SNAREs are present, it may stabilize the fully zipped state. A definitive evaluation of these and other models of Sec17 function will require the development of assays of the conformations of the C-terminal regions of SNARE domains in SNARE complexes, both is cis and in trans, with HOPS and with or without Sec17, and with complete or incomplete zippering.

Though C-terminal SNARE zippering is needed for fusion without Sec17/Sec18, it is unclear whether the remaining 3 C-terminal SNARE domains are disordered when Qc is C-terminally truncated by deletion of 21 amino acyl residues. Prior studies with incomplete zippering have suggested

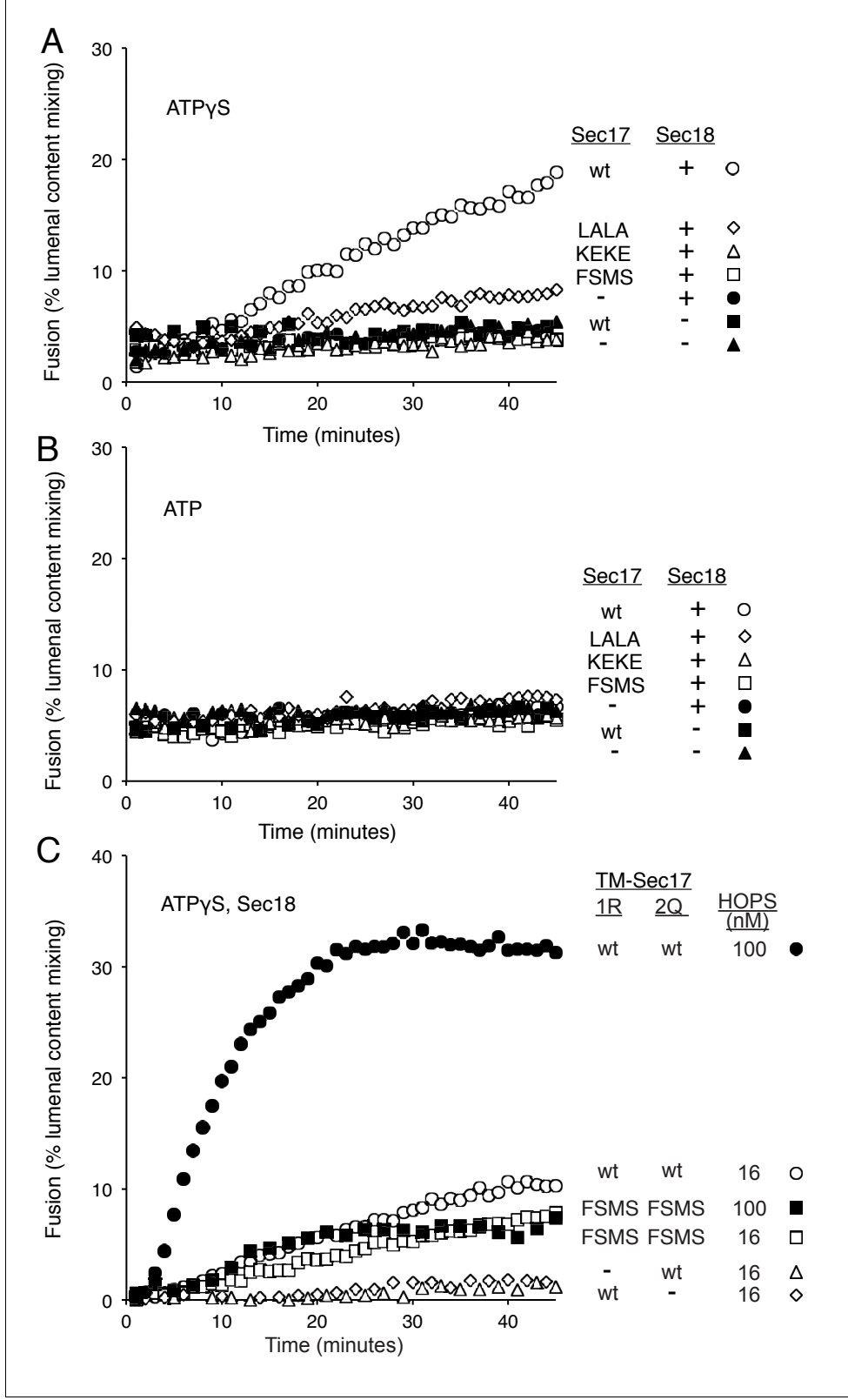

**Figure 6.** Fusion of proteoliposomes with Qc-3Δ. (**A**) Fusion reactions were performed with ATPγS as in *Figure 3A*, but with Qc-3Δ instead of Qc. (**B**) Fusion reactions were performed as in part A, but with ATP instead of ATPγS. Sec17 was wild type (wt) or had the mutations F21S,M22S (FSMS), K159E,K163E (KEKE), or L291A,L292A (LALA). (**C**) Fusion with Qc-3Δ needs the Sec17 apolar loop for more than membrane anchoring. Fusion with

*Figure 6 continued on next page*

*Figure 6 continued*

ATPγS and Qc-3Δ was with proteoliposomes bearing TM-Sec17 wild-type or TM-Sec17 F21S,M22S, and with 16 nM HOPS (open symbols) or 100 nM HOPS (filled symbols), as indicated.

The following source data and figure supplement are available for figure 6:

**Source data 1.** Source data file (Excel) for *Figure 6* Parts A and B.
**Source data 2.** Source data file (Excel) for *Figure 6C*.
**Figure supplement 1.** Average and standard deviations of fusion after 30 min for triplicate assays as in *Figure 6C*, relative to the maximal fusion condition in panels A and C, and as % lumenal compartment mixing in panel B.

either major conformational disorder of the remaining three helices, as determined by NMR in solution (*Trimbuch et al., 2014*), or that there is little effect on their structure, as seen when packed in a crystal (*Kümmel et al., 2011*). How do Sec17, Sec18, and ATPγS promote the fusion of proteoliposomes when SNARE zippering is arrested by the Qc-3Δ mutation? It is likely that ATPγS is

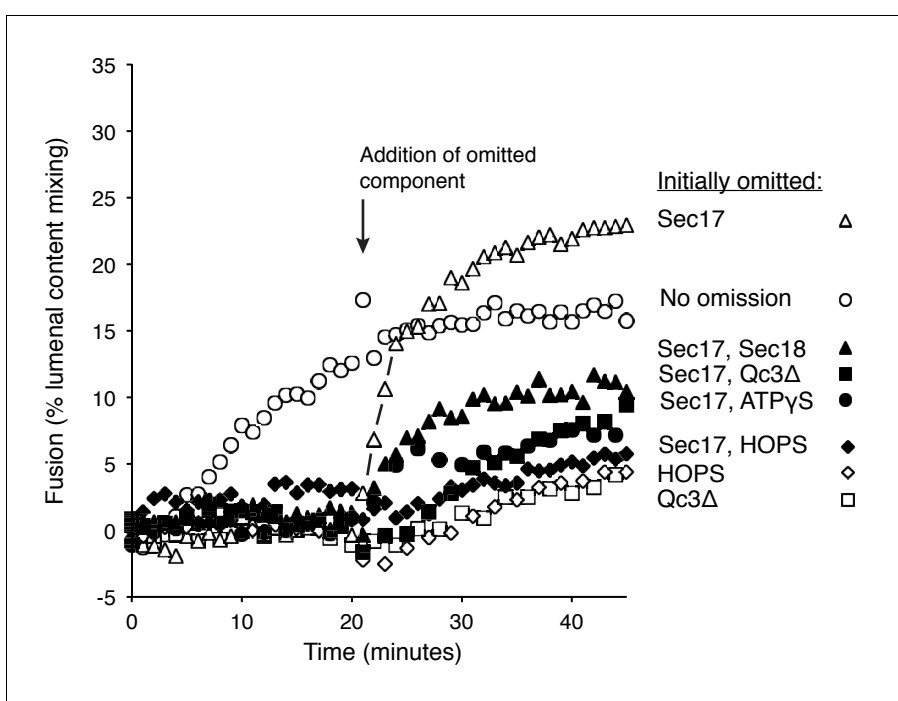

**Figure 7.** Assembly of an intermediate for rapid, Sec17-triggered fusion. Fusion incubations were performed with Ypt7:GTP, 1R-SNARE proteoliposomes and Ypt7:GTP, QaQb-SNARE proteoliposomes in the presence of HOPS, Sec17, Sec18, ATPγS, and Qc-3Δ as described in Materials and methods, with all components (open circles) or with omission of HOPS (open diamonds), Qc-3Δ (open squares), Sec17 (open triangles), Sec17 and Sec18 (filled triangles), Sec17 and Qc-3Δ (filled squares), or Sec17 and ATPγS (filled circles). After 20 min, the initial incubations (16 μl) were supplemented with 4 μl of Rb150 or of this buffer with the previously omitted components, so that each reaction then contained all the components at the same final concentrations: HOPS (31 nM), Sec17 (1.2 μM), Sec18 (72 nM), ATPγS (0.7 mM), and Qc-3Δ (16 nM).

The following source data and figure supplement are available for figure 7:

**Source data 1.** Source data file (Excel) for *Figure 7*.
**Figure supplement 1.** Average and standard deviations of the fusion rate for triplicate assays as in *Figure 7*.

functioning as an activating ligand to Sec18, and the sensitivity of fusion with Qc-3Δ to the Sec17-L291A,L292A mutation suggests that fusion stimulation requires Sec17:Sec18 direct association. In the 'trans-SNARE stability' model, the bound Sec17 would associate with the SNAREs to allow the three remaining C-terminal portions of the SNARE domains of the R-, Qa-, and Qb-SNAREs to be drawn together and to thereby draw the membranes more closely together, in effect substituting for the deleted Qc portion of the zipper. Optical tweezer studies (*Ma et al., 2016*) have shown that Sec17 (α-SNAP) binding can stabilize almost fully-zippered 4-SNARE bundles, a process of 'conformational selection,' rather than enhance the forward rate of zippering. Other studies using FRET have shown that the C-terminal domain of 4-SNARE complexes can be opened by α−SNAP (*Ryu et al., 2015*) or, in human synaptic model studies, by complexin (*Choi et al., 2016*). Even when the three remaining SNARE domain C-termini are gathered together by Sec17, they may not suffice for fusion, and the apolarity of the Sec17 N-domain loop may provide a wedge function to help initiate bilayer rearrangements. Indeed, with Qc-3Δ we find that both Sec18 and ATPγS are needed for the assembly of an intermediate which can drive a rapid fusion burst upon addition of Sec17 (*Figure 7*). With arrested SNARE zippering, Sec17 function relies on the apolar character of its loop in a manner which cannot be bypassed by an N-terminal trans-membrane anchor (*Figure 6C*).

In our working model of vacuole fusion, Sec18, Sec17, and ATP are necessary and sufficient for disassembly of *cis*-SNARE complexes on vacuoles or model proteoliposomes (*Mayer et al., 1996*; *Ungermann et al., 1998*; *Mima et al., 2008*). Ypt7:GTP on each membrane and HOPS, with its dual binding sites for Ypt7:GTP, suffice for tethering (*Hickey and Wickner, 2010*). Since Sec17 and Sec18 have been implicated in controlling tethering (*Mayer and Wickner, 1997*), the released Sec17 may modulate or displace HOPS from its SNARE associations (*Collins et al., 2005*), freeing it to interact with the Ypt7 Rab on each membrane for tethering. While HOPS and Rab tethering causes stable membrane association, it does not bring the two bilayer surfaces into immediate contact. A nexus of affinities among the proteins and lipids needed for fusion causes interdependent assembly of a ring-shaped fusion microdomain (*Wang et al., 2002*; ibid, 2003; *Fratti et al., 2004*), as recently also seen for mitochondria (*Brandt et al., 2016*). As part of the vacuolar fusion affinity nexus, the Rab has affinity for HOPS (*Hickey and Wickner, 2010*) and for the R-SNARE Nyv1 (*Orr et al., 2017*), HOPS has direct affinity for the R-, Qa-, and Qc- SNAREs (*Stroupe et al., 2006*; *Krämer and Ungermann, 2011*; *Lobingier and Merz, 2012*; *Baker et al., 2015*), the vacuolar SNAREs have affinity for each other (*Hickey and Wickner, 2010*), Sec17 and Sec18 can bind each other and to SNAREs (*Zhao et al., 2015*), and HOPS and the Qc-SNARE Vam7 bind PI(3)P and acidic lipids (*Stroupe et al., 2006*; *Lee et al., 2006*; *Karunakaran and Wickner, 2013*). Certain lipids such as diacylglycerol, sterol, and phosphoinositide are interdependent for enrichment in the fusion microdomain; strikingly, ligands to SNAREs can disrupt this lipid enrichment, and ligands to the lipids will prevent SNARE enrichment (*Fratti et al., 2004*). Once the components are enriched, HOPS catalyzes the assembly of SNAREs into a 4-SNARE complex in trans and may inhibit the disassembly of this trans-SNARE complex by Sec17/Sec18 (*Xu et al., 2010*) if it passes 'proofreading' criteria (*Starai et al., 2008*; *Lobingier et al., 2014*). The SNARE domains then zipper in the N to C direction. After substantial zippering, there is coordinated binding and function of Sec18 and several Sec17 monomers. The insertion of the Sec17 apolar N-domain loops into the adjacent bilayers may perform two functions, triggering rapid lipid rearrangement for fusion, and/or stabilizing the Sec17 membrane-bound state so that this bound Sec17 can stabilize the otherwise reversible and labile fully-zippered state of the *trans*-SNARE complex. Optimal fusion rates with wild-type SNAREs require both Sec17 and Sec18 and complete, C-terminal SNARE zippering (*Figures 2A* and *5*). ATP or, in reconstituted systems, ATPγS serves as an activating Sec18 ligand, but ATP hydrolysis is not required to promote fusion (*Figure 2A*). After fusion, Sec18 uses the energy of ATP hydrolysis to disassemble cis-SNARE complexes (*Figures 2B* and *3B*), releasing Sec17 (*Mayer et al., 1996*) and the soluble Qc SNARE Vam7 (*Boeddinghaus et al., 2002*). Further studies will be required to determine whether the same molecules of Sec17, Sec18, and ATP which engage t*rans*-SNARE complexes to enhance fusion remain bound after fusion converts them to *cis*-SNARE complexes, which HOPS no longer protects from Sec18/Sec17-mediated disassembly (*Xu et al., 2010*).

Our working model is a useful basis for experiments, but substantial puzzles remain. How does Sec18 alone, with bound ATP or ATPγS, stimulate fusion (*Figure 2A*)? The structure of the NSF (Sec18)/α-SNAP(Sec17)/4-SNARE '20 s' complex (*Zhao et al., 2015*) shows Sec18 binding to the 4-SNARE coiled coil near the N-terminus of the SNARE domains, quite removed from the bilayer. We

note that Sec18 can disassemble SNARE complexes for fusion in the absence of Sec17, albeit at greatly reduced rates (*Zick et al., 2015*). Why is ATPγS, but not ATP, able to support HOPS-, Sec17- and Sec18-dependent fusion when SNAREs cannot complete zippering? Perhaps partially-zippered SNARE complexes are more prone to Sec18/ATP mediated disassembly than fully zippered ones, but this is only a hypothesis.

How do the current findings relate to earlier results? In an earlier study, asking whether Sec17 affects fusion in the absence of Sec18, we employed proteoliposomes bearing either the R- or Qa-SNARE at high levels (1:2000 molar ratio to lipids) and incubated them with HOPS, the soluble Qc-SNARE, and a mutant form of the Qb-SNARE which was rendered soluble by deletion of its trans-membrane anchor (*Zick et al., 2015*). Under these conditions, where none of the SNAREs were initially in complexes, we presumed that no Sec18 would be needed. When these liposomes were prepared with the full vacuolar mixed lipids of physiological fluidity, there was substantial fusion with HOPS alone and only minor stimulation by Sec17. To see strong Sec17-dependence, fusion had to be further suppressed by nonphysiological head-group or fatty acyl compositions. There was no further stimulation by Sec18, alone or with ATP, but Sec18 and ATPγS did stimulate. We have now asked whether Sec17 and Sec18 can promote fusion with wild-type SNAREs at physiological concentrations (1:32,000 molar ratio to lipids) and physiological lipid head-groups and fatty acyl fluidity (*Zick and Wickner, 2016*). We find that the completion of SNARE-zippering, the apolar loop of Sec17, and its interactions with the SNAREs will promote fusion.

In earlier studies of the fusion of isolated vacuoles, we reported (*Mayer et al., 1996*) that Sec17/Sec18 are only required early in the fusion pathway, supporting *cis*-SNARE complex disassembly. Under those in vitro reaction conditions, the substantial lag between docking and fusion might have allowed complete SNARE zippering, obscuring any contribution by Sec17 and its apolar loop to the fusion rate.

## Materials and methods

Proteins originating in *S. cerevisiae* (RRID: NCBITaxon_4932) for proteoliposome preparation and for the fusion assay were prepared as described in *Zick and Wickner (2016)* and stored at −80°C in small aliquots after freezing in liquid nitrogen. Lipids were from Avanti (Alabaster, AL), though ergosterol was from Sigma (St Louis, MO), PtdIns(3)P was from Echelon (Salt Lake City, UT), and Marina blue-PE and NBD-DHPE were from Thermo Fisher (Waltham, MA). Cy5-streptavidin was from SeraCare (Milford, MA) and biotinylated and underivatized streptavidin were from Thermo Fisher. Protein was assayed by the method of *Bradford (1976)*.

### Proteoliposome preparation

Proteoliposomes were prepared as described (*Zick and Wickner, 2016*). For lipid stocks, chloroform solutions of lipids which had been stored at −70°C were warmed to room temperature for 30 min, nutated for 30 min, then vortexed for 10 s. Lipid stocks were delivered with Hamilton syringes into glass vials. The vials received 450 µl of 0.5M β-octylglucoside in methanol, 268 µl of 25 mg/ml 18:2 phosphatidylcholine, 96 µl of 25 mg/ml 18:2 phosphatidylethanolamine, 112 µl of 25 mg/ml soy phosphatidylinositol, 26 µl of 25 mg/ml 18:2 phosphatidylserine, 9.9 µl of 18:2 phosphatidic acid, 117 µl of 5 mg/ml ergosterol, 52 µl of 2 mg/ml diacylglycerol, 173 µl of 1 mg/ml PtdIns(3)P, and either (A) 34 µl of 1 mg/ml Marina Blue-phosphatidylethanolamine or (B) 103.5 µl of 5 mg/ml NBD-phosphatidylethanolamine. Portions, 297 µl of A and 309 µl of B, were transferred to 2 ml glass vials and solvent removed under a stream of nitrogen for 30 min. Vials were then transferred to a Speed-Vac to remove any remaining solvent under vacuum for at least 3 hr at room temperature. Each vial then received 0.4 ml of 50 mM HEPES/NaOH, pH 7.4, 0.375M NaCl, 25% (v/v) glycerol, 2.5 mM MgCl$_2$. After sealing with Teflon-lined caps, vials were nutated with the lipidic pellet facing down for 3 hr. Vials were stored at −80°C, then thawed and nutated for 30 min at room temperature.

Protein stocks for Ypt7/1R and Ypt7/2Q proteoliposome preparation were as follows: Mix 1R; 7.47 µl of 59 µM Nyv1 were mixed with 692.6 µl of Rb150 (20 mM HEPES/NaOH, pH 7.4, 0.15M NaCl, 10% (v/v) glycerol) with 1% β-octylglucoside. Mix 2Q; 8.11 µl of 33.6 µM GST-TEV-Vam3, 10.89 µl of 40 µM Vti1, and 676 µl of Rb150 plus 1% β-octylglucoside. Mix Ypt7; 157 µl of Rb150 plus 1% β-octylglucoside, 526.4 µl of Ypt7 buffer (20 mM TrisCl, pH 8, 50 mM NaCl, 1 mM MgCl$_2$, 0.5% β-octylglucoside, 20 mM reduced glutathione), 195.8 µl of 16.6 µM prenylated Ypt7, 95.2 µl of

68 µM TEV protease. Cy5-labeled streptavidin (1 mg, from SeraCare) was dissolved in 592 µl water, and 21 µl water was added to 0.5 ml of 4 mg/ml biotinylated phycoerythrin (Thermo Fisher).

Solutions for dialysis were prepared on ice in the vials with lipid/detergent solutions. Biotin-phycoerythrin (250 µl) was added to each vial A and 250 µl Cy5-streptavidin to each vial B, then 200 µl of 1R was added to each A, and 200 µl 2Q was added to each B. Ypt7 solution (150 µl) was added to each vial. Vails were closed with Teflon-sealed screw caps, inverted three times, and nutated for 30 min at 4°C. Each 1 ml mixed micellar solution was then loaded into 6 cm of dialysis tubing (diameter 7.5 mm, Spectrum Labs, Rancho Dominiguez, CA, rinsed with water and knotted at one end), drawn taut and knotted at the other end, then dialyzed vs. 250 ml of Rb150 +1 mM MgCl$_2$ with 1 g BioBeads-SM2 (BioRad, Hercules, CA) for 18 hr at 4°C with vigorous stirring in the dark. Multiple bags were often dialyzed in the same vessel with proportional volumes of Rb150 + MgCl$_2$ and BioBeads.

Dialyzed proteoliposomes were harvested, floated through step gradients of Histodenz, harvested from the step gradients, assayed for lipid phosphorus, diluted with Rb150 + Mg to a final concentration of 2 mM, aliquoted in 30 µl portions into strips of 250 µl snap-cap tubes, frozen in liquid N$_2$, and stored at −80°C as described (*Zick and Wickner, 2013*).

## TM-anchored Sec17 and Sec17-F21S,M22S

The DNA encoding the transmembrane (TM) domain from the Qb-SNARE Vti1 (Amino acid 192–217) was joined to the cDNA encoding GST-TEV-fused Sec17 (WT) or Sec17 (F21S,M22S) to yield GST-TEV-TM-Sec17 (WT) and GST-TEV-TM-Sec17 (FSMS). Constructs were prepared with recombinase in-Fusion (Clonetech). The TM domain of Vti1 and Sec17 (WT) or Sec17 (FSMS) DNA fragments were PCR-amplified using the primers below and cloned into pGST-parallel1 vectors (*Sheffield et al., 1999*), encoding GST and a TEV protease site.

For Transmembrane domain of Vti1
F: AG GGC GCC ATG GAT CCG GCTAATAAATTCATAAGCTAT
R: TTTAAACTTTGAGAACAAAACTAGC
For Sec17 (WT) or Sec17 (FSMS)
F: TGTTCTCAAAGTTTAAA ATGTCAGACCCTGTAGAGTTA
R: AGTTGAGCTCGTCGA TCATAACAAATCATCTTCTTG

## Protein expression and purification

GST-TEV-TM-Sec17 (WT) or GST-TEV-TM-Sec17(FSMS) were produced in E.coli Rosetta pLysS (Novagen). Transformants were grown in 3L of LB media with ampicillin and chloramphenicol at 37°C to an absorbance at 600 nm of 1, induced with 1 mM IPTG, then grown overnight (10 hr, 18°C, 4 L). Cells were harvested and suspended in 40 ml resuspension buffer (20 mM TrisCl, pH 8.0, 200 mM NaCl, 1 mM PMSF and PIC (protease inhibitor cocktail; *Xu and Wickner, 1996*), lysed by two passages in a French Press at 4°C, and centrifuged (60Ti Beckman), 50,000 rpm, 1 hr, 4°C). Lysate pellets were resuspended in 30 ml of extraction buffer (20 mM TrisCl, pH 8.0, 150 mM NaCl, 1 mM EDTA, 1 mM DTT, 1% Triton X100, PIC and 1 mM PMSF) with a Dounce homogenizer and incubated (4°C, 1 hr) with gentle agitation.

The extract was centrifuged (60Ti rotor, 50,000 rpm, 1 hr, 4°C) and the supernatant was applied to a 1.5 × 5 cm glutathione-Sepharose column in extraction buffer. The column was washed with 100 ml of 20 mM HEPES-NaOH, pH 7.5, 150 mM NaCl, 1 mM EDTA, 1 mM dithiothreitol, and 100 mM β-octylglucoside. GST-TEV-TM-Sec17 was eluted with 20 mM Hepes-NaOH, pH 7.5, 150 mM NaCl, 1 mM EDTA, 1 mM DTT,10% glycerol, 100 mM β-octylglucoside and 20 mM reduced glutathione. The GST tag was removed by the TEV present during proteoliposome formation (above).

## Assay of fusion

Ypt7/1R- and Ypt7/2Q- proteoliposomes (final concentration 0.55 mM lipid each) were incubated together for 10 min at 27°C with 10 µM nonfluorescent streptavidin, 1 mM EDTA, and 70 µM GTP, then returned to ice and MgCl$_2$ was added to 2.64 mM. These proteoliposomes, with GTP-loaded Ypt7, and a separate mixture of 32 nM Vam7, 0.7 mM ATP or ATPγS, 660 nM Sec17, 334 nM Sec18, and 64 nM HOPS (except as noted in the Figure Legends) were incubated for 10 min at 27°C in wells of a 384-well plate in a fluorescence plate reader before 10 µl of each was mixed with a multichannel

micropipettor. FRET between lumenal Cy5-streptavidin and biotin-phycoerythrin was assayed each minute in each sample, as described (*Zick and Wickner, 2016*).

## Acknowledgements

This work was supported by NIH grants R01 GM23377-40 and R35GM118037-01 to WW and GM077349 to AM. We thank Michael Zick, Charles Barlowe, and Gustav Lienhard for fruitful discussions and Deborah Douville for expert assistance.

## Additional information

### Funding

| Funder | Grant reference number | Author |
| --- | --- | --- |
| National Institutes of Health | R01 GM23377-40 | William Wickner |
| National Institutes of Health | R35GM118037-01 | William Wickner |
| National Institutes of Health | GM077349 | Alexey J Merz |

The funders had no role in study design, data collection and interpretation, or the decision to submit the work for publication.

### Author contributions

HS, Conceptualization, Data curation, Formal analysis, Investigation, Methodology, Writing—original draft, Writing—review and editing; AO, Data curation, Formal analysis, Investigation, Methodology, Writing—review and editing; MD, Investigation, Methodology; AJM, WW, Conceptualization, Data curation, Formal analysis, Funding acquisition, Investigation, Methodology, Writing—original draft, Writing—review and editing

### Author ORCIDs

Alexey J Merz, http://orcid.org/0000-0003-2177-6492
William Wickner, http://orcid.org/0000-0001-8431-0468

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
