## [Decision Letter]

Thank you for submitting your article "Sec17/Sec18 act twice, triggering fusion independent of full SNARE zippering and then disassembling cis-SNARE complexes" for consideration by *eLife*. Your article has been reviewed by three peer reviewers, one of whom is a member of our Board of Reviewing Editors and Randy Schekman as the Senior Editor. The following individual involved in review of your submission has agreed to reveal his identity: Josep Rizo (Reviewer #2).

The reviewers have discussed the reviews with one another and the Reviewing Editor has drafted this decision to help you prepare a revised submission.

Summary:

This is a very interesting paper that is at the same time paradoxical, because it was the Wickner lab that in 1996 brought the notion that Sec18 acts before membrane fusion, rather than being the fusion protein as had been proposed at the time, and now the Wickner lab brings back Sec18 to action at the fusion step. Here a new function of Sec17 and Sec18 is discovered and these factors can bypass a defect of a C-terminal deletion mutant (Qc-3Delta) of the soluble Qc-SNARE Vam7 involved in vacuolar fusion. This new function occurs in the presence of ATPgS, so it does not require ATP hydrolysis.

Previously, Schwartz and Merz (2009) found that Sec17 can partially rescue the defect of the Qc-3Delta mutant, but these experiments were performed at relatively high SNARE protein concentration, and the rescue was not complete. In the present work, it is shown that Sec17/Sec18/ATP (or ATPgS) produce comparable fusion rates for Qc-3Delta compared to the wildtype situation (under physiological conditions with relatively low SNARE protein concentrations and lipid compositions). The authors conclude that Sec17 and Sec18 have a function at the assembly stage of trans SNARE complex formation in addition to their established role of disassembly of SNARE complexes.

These results are important for the field of intracellular membrane fusion and will spark many new experiments by other labs.

Essential revisions:

1) In the context of Figure 5, it would be interesting to also probe the interactions found by Zhao et al., (2015) that affect the interaction between Sec17 and the SNARE complex, in particular mutations that interact with SNARE residues near the ionic layer and that disrupt disassembly function (Figure 6 in Zhao et al., 2015). Such an experiment would support the proposed notion that it is the interaction between Sec17 and the SNARE complex that compensates for the deletion of the C-terminal residues of Vam7.

2) In Figure 2, it would be interesting to conduct an experiment with wildtype Vam7 and with one of the Sec17 mutants that disrupts rescue of the Qc-3Delta phenotype in the presence of ATPgS but only mildly reduces disassembly (i.e., the mutation of the membrane binding loop of Sec17). Such an experiment might show more directly that the presence of Sec17/Sec18/ATPgS and the interaction with the membrane is stimulatory in the case of wildtype Vam7.

3) In Figure 5, the authors tested the Sec17-L291AL292A and Sec17-F21SM22S mutation, altering the interaction with Sec18 and interaction with the membrane, respectively and found them to be essential for the rescue of Qc-3Delta. Of particular note is that the membrane interaction of Sec17 is essential for the rescue function of the Qc-3Delta mutant, but this mutant does not entirely disrupt the disassembly function of NSF/aSNAP with or without liposomes in vitro (Winter et al., JBC, 2009). This point could be discussed in more detail since it is relevant to the one of the possible models proposed by the authors, i.e., that Sec17 stabilizes the partially truncated trans SNARE complex with Qc-3Delta.

4) In the Discussion: The work by Ryu et al., (2015) suggests that the C-terminal part of the SNARE complex can be opened by aSNAP. Please note that this was studied using the soluble fragment of synaptobrevin, i.e., there was only one membrane anchor. Choi et al., (2016) discovered that complexin can also open the C-terminal part of the SNARE complex, but again only one membrane anchor was present in these experiments. Thus, in the context of full-length SNAREs (i.e., with both synaptobrevin and syntaxin transmembrane domains included), such an intricate interaction between complexin or aSNAP probably does not occur for the cis-SNARE complex, but, rather, it should occur for the trans SNARE complex as proposed by Choi et al., (2016). Thus, the available data from all these experiments are consistent with an interaction between complexin or aSNAP with the trans SNARE complex. Some discussion of these points might be appropriate.

5) To derive the alternative fusion pathway, the authors assumed that the vacuolar SNARE complex containing truncated Qc does not fully zipper. However, the authors did not provide any evidence supporting this assumption. Truncating the C-terminus of Qc likely weakens SNARE zippering, but does not necessarily abolish full SNARE zippering. Actually, this is exactly what one of the coauthors of this manuscript proposed earlier (Schwartz and Merz 2009). For neuronal SNARE complexes, although NMR data showed that truncations in the C-terminus of one SNARE motif render the C-terminal regions of the other SNARE motifs unstructured (Trimbuch et al., 2014), crystal structures showed that truncating the C-terminus of the R SNARE does not significantly perturb the conformations of the other three helices in the SNARE bundle (Kummel D. et al., 2011). Hence, much like crystal packing stabilized the C-terminal half of the complex in this case, interactions with other proteins such as Sec17 can also provide such stabilization and thus favor full zippering of the other three helices. A discussion of these points would be desirable.

6) A possible alternative model is that the energy loss due to SNARE mutation or truncation can be compensated by enhancing the role of regulatory proteins in membrane fusion. For example, cleavage of the C-terminal 9 amino acids of the Qc SNARE motif in SNAP-25 by BoNT A attenuates neurotransmitter release. But neurotransmitter release is rescued by evaluated calcium concentration, presumably due to enhanced synaptotagmin binding to membranes. Sec17 or α-SNARE has been shown to enhance SNARE zippering and promote membrane fusion. Thus, it is likely that Sec18 helps to recruit Sec17 to SNAREs and further enhance full SNARE zippering, which compensates the energy loss due to Qc truncation. This interpretation is consistent with the requirements of ATPγS and membrane binding of Sec17 through its N-terminus. It is known that ATPγS helps to form the 20S NSF/SNAP/SNARE complex. Thus, the multivalent interactions facilitate recruitment of Sec17 to SNAREs, which enhance full SNARE zippering and membrane fusion. To enhance SNARE zippering, Sec17 does not need to bind to the central hydrophobic core of the other three helices in a way similar to the missing Qc C-terminus. Instead, any binding to the zippered helixes, even to their surfaces, suffices to enhance SNARE zippering. Again, more discussion would be desirable.

---

## [Author Response]

*Essential revisions:*

*1) In the context of Figure 5, it would be interesting to also probe the interactions found by Zhao et al., (2015) that affect the interaction between Sec17 and the SNARE complex, in particular mutations that interact with SNARE residues near the ionic layer and that disrupt disassembly function (Figure 6 in Zhao et al., 2015). Such an experiment would support the proposed notion that it is the interaction between Sec17 and the SNARE complex that compensates for the deletion of the C-terminal residues of Vam7.*

We have prepared this mutant, characterized its diminished affinity for SNAREs in the new Figure 3 Supplement, and characterized its effects on fusion with ATP or ATPgS, and with wild-type Vam7 or Vam7-3D, in the new Figure 3 and 6. Thank you for suggesting this. We’ve gone well beyond doing just what was requested, and now present a more systematic view.

*2) In Figure 2, it would be interesting to conduct an experiment with wildtype Vam7 and with one of the Sec17 mutants that disrupts rescue of the Qc-3Delta phenotype in the presence of ATPgS but only mildly reduces disassembly (i.e, the mutation of the membrane binding loop of Sec17). Such an experiment might show more directly that the presence of Sec17/Sec18/ATPgS and the interaction with the membrane is stimulatory in the case of wildtype Vam7.*

As noted above, we now systematically compare fusion with ATP vs ATPgS, with wild type Vam7 vs Vam7-3D, and with Sec17 that’s either wild-type or has lost the apolarity of its N-domain loop, has diminished affinity for the SNARE 0-layer, or has altered residues in its C-terminal domain which modulate its Sec18 interactions.

To address critically the function of Sec17’s N-domain apolar loop, we also prepared membrane anchored forms of Sec17 that are either (otherwise) wild-type or bear the F22SM23S mutation. We find (Figure 4) that the FSMS mutation has little effect with membrane-anchored Sec17 on fusion with wild-type Vam7 (Figure 4 vs E-G), in contrast to its strong effect when Sec17 is soluble (Figure 3). This indicates that the primary function of the apolar loop is not as a “wedge” but to promote Sec17 membrane binding (since it can be bypassed by the addition of a trans-membrane domain). In contrast, when completion of SNARE zippering is blocked, addition of a membrane anchor to Sec17 does not bypass the need for the apolarity of its apolar loop (Figure 6).

*3) In Figure 5, the authors tested the Sec17-L291AL292A and Sec17-F21SM22S mutation, altering the interaction with Sec18 and interaction with the membrane, respectively and found them to be essential for the rescue of Qc-3Delta. Of particular note is that the membrane interaction of Sec17 is essential for the rescue function of the Qc-3Delta mutant, but this mutant does not entirely disrupt the disassembly function of NSF/aSNAP with or without liposomes* in vitro *(Winter et al., JBC, 2009). This point could be discussed in more detail since it is relevant to the one of the possible models proposed by the authors, i.e., that Sec17 stabilizes the partially truncated trans SNARE complex with Qc-3Delta.*

We have extensively revised the Discussion, and agree that the Winter et al., data is of direct relevance to our work and is therefore part of the Discussion.

*4) In the Discussion: The work by Ryu et al., (2015) suggests that the C-terminal part of the SNARE complex can be opened by aSNAP. Please note that this was studied using the soluble fragment of synaptobrevin, i.e., there was only one membrane anchor. Choi et al., (2016) discovered that complexin can also open the C-terminal part of the SNARE complex, but again only one membrane anchor was present in these experiments. Thus, in the context of full-length SNAREs (i.e., with both synaptobrevin and syntaxin transmembrane domains included), such an intricate interaction between complexin or aSNAP probably does not occur for the cis-SNARE complex, but, rather, it should occur for the trans SNARE complex as proposed by Choi et al., (2016). Thus, the available data from all these experiments are consistent with an interaction between complexin or aSNAP with the trans SNARE complex. Some discussion of these points might be appropriate.*

While we respectfully wish to minimize our discussion of complexin, since it’s not in the vacuolar system and is itself the topic of intense current study and some controversy, but we have expanded and hopefully clarified that aSNAP has been shown to bind the C-terminal regions of SNAREs, especially in the trans-complex. We’d welcome any further suggestions for strengthening this part of the Discussion.

*5) To derive the alternative fusion pathway, the authors assumed that the vacuolar SNARE complex containing truncated Qc does not fully zipper. However, the authors did not provide any evidence supporting this assumption. Truncating the C-terminus of Qc likely weakens SNARE zippering, but does not necessarily abolish full SNARE zippering. Actually, this is exactly what one of the coauthors of this manuscript proposed earlier (Schwartz and Merz, 2009). For neuronal SNARE complexes, although NMR data showed that truncations in the C-terminus of one SNARE motif render the C-terminal regions of the other SNARE motifs unstructured (Trimbuch et al., 2014), crystal structures showed that truncating the C-terminus of the R SNARE does not significantly perturb the conformations of the other three helices in the SNARE bundle (Kummel D. et al., 2011). Hence, much like crystal packing stabilized the C-terminal half of the complex in this case, interactions with other proteins such as Sec17 can also provide such stabilization and thus favor full zippering of the other three helices. A discussion of these points would be desirable.*

We now explicitly state that we don’t know the conformation of the remaining 3 SNARE C-terminal domains, and further discuss this point, including these very papers in the Discussion. What we call the “trans-SNARE stability” model could have Sec17 interacting with the C-terminal regions of SNAREs; as stated in the Discussion now, when Qc-3D is present Sec17 might stabilize the remaining 3 SNARE C-terminal regions in a conformation that resembles the wild-type fully zipped conformation, and when the wild-type full-length SNAREs are present, it may interact to stabilize the fully zipped state.

*6) A possible alternative model is that the energy loss due to SNARE mutation or truncation can be compensated by enhancing the role of regulatory proteins in membrane fusion. For example, cleavage of the C-terminal 9 amino acids of the Qc SNARE motif in SNAP-25 by BoNT A attenuates neurotransmitter release. But neurotransmitter release is rescued by evaluated calcium concentration, presumably due to enhanced synaptotagmin binding to membranes. Sec17 or α-SNARE has been shown to enhance SNARE zippering and promote membrane fusion. Thus, it is likely that Sec18 helps to recruit Sec17 to SNAREs and further enhance full SNARE zippering, which compensates the energy loss due to Qc truncation. This interpretation is consistent with the requirements of ATPγS and membrane binding of Sec17 through its N-terminus. It is known that ATPγS helps to form the 20S NSF/SNAP/SNARE complex. Thus, the multivalent interactions facilitate recruitment of Sec17 to SNAREs, which enhance full SNARE zippering and membrane fusion. To enhance SNARE zippering, Sec17 does not need to bind to the central hydrophobic core of the other three helices in a way similar to the missing Qc C-terminus. Instead, any binding to the zippered helixes, even to their surfaces, suffices to enhance SNARE zippering. Again, more discussion would be desirable.*

This is just the consideration we’re now making more explicit in the Discussion as the “trans-SNARE stability” model.